# Sampling Parallel SOA-MZIs Configuration for All-Optical Simultaneous Frequency Down-Conversion

**Hassan Termos [1],\* and Ali Mansour [2]**

1    ICFO—The Institute of Photonic Sciences, 08860 Castelldefels, Barcelona, Spain
2    Lab-STICC, CNRS UMR 6285, ENSTA Bretagne, 2 Rue François Verny, CEDEX 09, 29806 Brest, France
\*    Correspondence: hassan.termos@icfo.eu

**Abstract:** In this paper, we expound a modulation concept to contrive simultaneous frequency down-conversion based on a three parallel Semiconductor Optical Amplifier Mach-Zehnder Interferometers (SOA-MZIs) link by using a band pass sampling method in a Virtual Photonics Inc. simulator. Each SOA-MZI is deployed to achieve a down-converted signal, which has ten replicas related to the first ten harmonic ranks of the sampling signal, at the SOA-MZI outer port. Then, the admixture of the three down-converted signals yields a sampled signal, which is called a simultaneous down-converted signal that contains thirty different replicas. The positive down-conversion gains with top values are reached with the sampling parallel SOA-MZIs link. Moreover, we evaluated the quality of the parallel SOA-MZIs transmission system over orthogonal frequency division multiplexing (OFDM) complex modulated signals using the error vector magnitude values as a performance index. The utmost bit rate attained is 2 Gbit/s for OFDM modulations.

**Keywords:** all-optical sampling mixer; frequency down-conversion; parallel semiconductor optical amplifier Mach-Zehnder interferometers; orthogonal frequency division multiplexing





## 1. Introduction

The sampling photonic mixers play a significant part in radio over fiber (RoF) systems and millimetre-wave applications. Recently, all-optical mixers have been widely considered thanks to many advantages, such as: wide transmission band, low power loss, and low cost [1] compared to conventional electrical mixers. Semiconductor Optical Amplifier Mach-Zehnder Interferometers (SOA-MZIs), which exploit cross gain modulation (XGM) and cross phase modulation (XPM) [2], are utilized for frequency mixing. SOA-MZIs provide a large extinction ratio, low optical input optical power, and positive conversion gains (CGs) [2]. The parallel SOA-MZIs concept [3–6], which has crucial characteristics, is also used for simultaneous frequency down-conversion by applying a band pass sampling method.

Recently, experimental and simulated systems of frequency down-conversion principle relied on a single SOA-MZI or a cascaded SOA-MZIs link [7,8] exhibited good performances through error vector magnitudes (EVMs) and conversion gains (GCs). Hereinafter and for the first time, we design a parallel sampling SOA-MZIs system to contrive simultaneous frequency down-conversion of three radiofrequency (RF) signals based on a standard modulation configuration by using a Virtual Photonics Inc. (VPI) simulator. The CGs are measured for three down-conversion ranges from 98 to 0.5 GHz, 98.5 to 1 GHz, and 99 to 1.5 GHz. The highest CG is reached for a sampled signal from 98 GHz to 0.5 GHz. Moreover, we examine the quality of the sampling parallel SOA-MZIs system over orthogonal frequency division multiplexing (OFDM) complex modulated signals in order to identify its unprecedented performance in terms of bit error rate (BER) and EVM values. This optical transmission system is a novel principle dependent on a sampling parallel SOA-MZIs link to realize simultaneous down-conversion that includes thirty distinct replicas.

## 2. Principle of Simultaneous Down-Conversion Based on a SOA-MZIs Link

The two parallel SOA-MZIs were recognized for wavelength conversion in [3–6]. Based on a three-parallel SOA-MZIs link, the major novelty of our system consists of achieving a simultaneous frequency down-conversion by using a sampling method. The SOA-MZI structure in the Virtual Photonics Inc. (VPI) simulator and its static and dynamic behaviours are expounded in [8].

The promising optical system of a sampling parallel SOA-MZIs link relies on the cross-phase modulation (XPM) effect of the two input signals for each SOA-MZI upper arm as displayed in Figure 1. In this schema, for SOA-MZI1, the incoming data signal at the frequency $f_1$ is entered in SOA1 through the upper port (UP) where it modulates the gain of SOA1, and thereby the gain and phase of the sampling signal. In other words, the sampling signal is modulated by the data RF signal. This sampling signal with a sampling frequency $f_s$ at a wavelength $\lambda_s$ is coupled into the middle port (MP) and it is split into upper and lower arms at the same time by using an optical coupler (OC). A phase shift only occurs on the sampling signal at the upper arm influenced by the data signal. Therefore, this gives rise to a phase modulation of the sampling signal propagating in SOA1 according to the data signal at the UP input. In the lower arm, the sampling signal is only amplified.

At the SOA-MZI1 output, the signals from the two SOAs interfere in order to generate the sampled signal that corresponds to the down-converted signal. The quality of this latter signal is upgraded due to enhancing the sampling signal that maintains the same electric power of its harmonics. As displayed in the electrical spectrum, Figure 1A, of the sampled signal, the data signal is down-converted from $f_1$ to mixing frequencies $|f_1 - nf_s|$ due to the band pass sampling method [9], where $n$ ranges from 1 to 10 in this principle in Figure 1.

On the other hand, the same operation is also accomplished for the second and third SOA-MZIs. So, the second radio frequency (RF) signal is down-converted from $f_2$ to $f_2 - nf_s$ at the SOA-MZI2 output as illustrated in Figure 1B and the third one is down-converted from $f_3$ to $f_3 - nf_s$ at the SOA-MZI3 outer as displayed in Figure 1C. As a result, these down-converted signals at the SOA-MZIs outputs are combined by an optical coupler (OC) in order to achieve simultaneous frequency down-conversion of distinct replicas at $|f_m - nf_s|$ as propounded in Figure 1D.

As propounded in Figure 1D, the replicas at $|f_m - nf_s|$, $m \in \{1, 2, 3\}$ have green, dark blue, or purple colours corresponding to the sampled signals at the output of each SOA-MZI after combination in order to achieve simultaneous frequency mixing. The RF frequencies at $f_m$ are amplified as presented in Figure 1D at the SOA-MZIs output. First, $f_1$ in blue colour corresponds to the RF frequency related to the first data signal presented at the upper port (UP) input of the SOA-MZI1. Next, $f_2$ in light blue colour corresponds to the RF frequency related to the second data signal at the UP input of the SOA-MZI2. Finally, $f_3$ in orange colour correlates with the RF frequency of the third data signal at the UP input of the SOA-MZI3. The harmonics of the sampling signal at $nf_s$ are shown in red. These harmonics have distinct amplitudes because they decrease inversely to their frequencies. This results in replicas with different amplitudes relaying on the harmonic of the sampling signal involved for a particular replica.

It is worth noting that the parameter $n$ should be an integer that stands for the harmonic rank of the sampling signal. The sampled signal follows these harmonics dependent on their positions at $nf_s$. In this paper, $n$ is limited to 10 because we want to achieve the down-conversion process based on the sampling parallel SOA-MZIs link. Hence, the up-conversion process can also be recognized when $n > 10$. This is because the mixing frequencies of replicas related to harmonics at $nf_s$ where $n \geq 11$ are higher than the RF frequencies at $f_m$. In Figure 1A–C, we can notice ten replicas of the sampled signal; so mentioning the formula for every target frequency may increase the provided details in the figure and confuse the readers. Therefore, relevant formulas have been added.

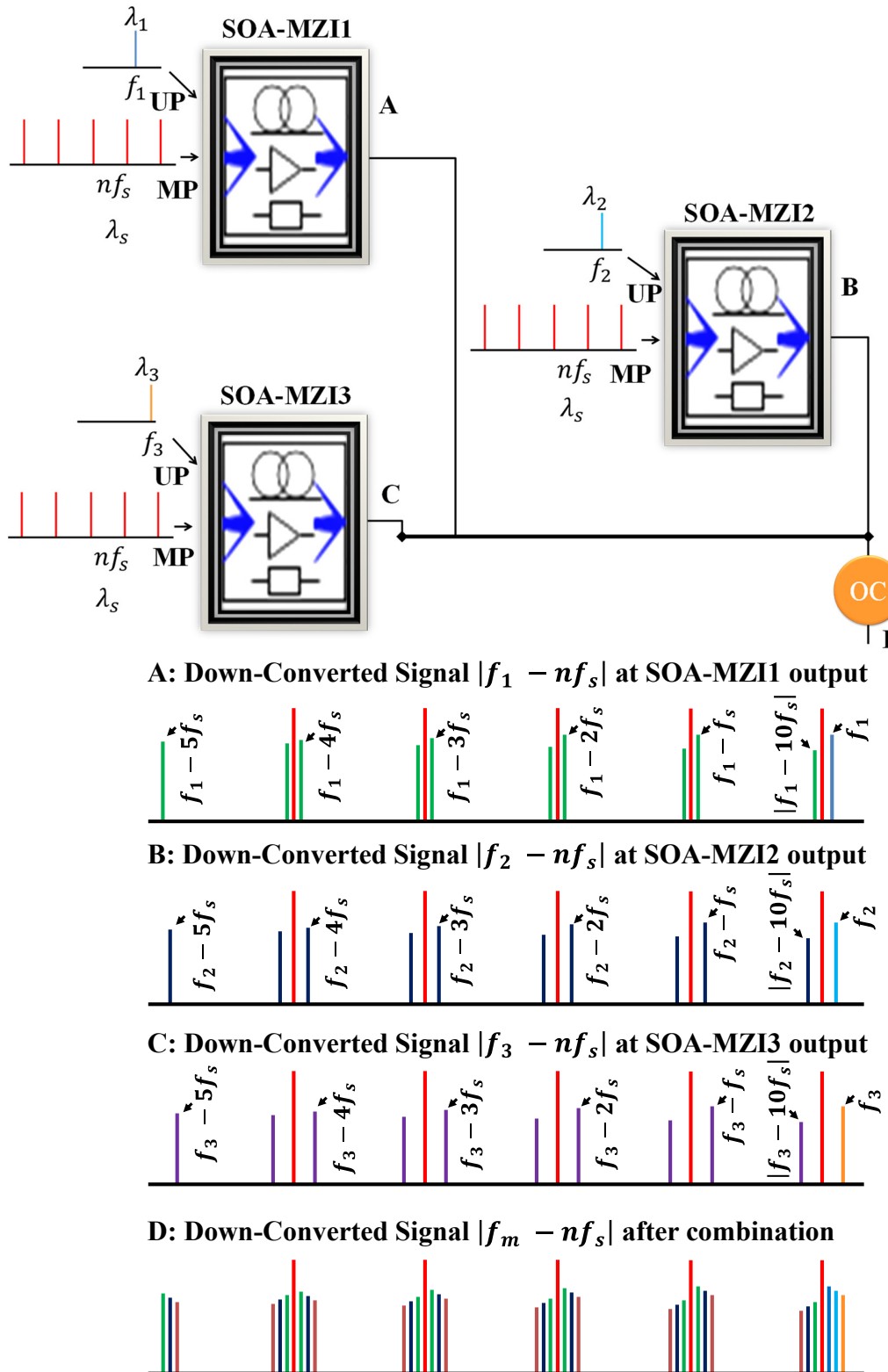

**Figure 1.** Frequency Down-conversion principle based on the sampling parallel SOA-MZIs in the standard modulation schema. OC: Optical Coupler, $f_1$, $f_2$, and $f_3$: frequency of the first, second, and third data RF signal, respectively, $f_s$: sampling frequency, UP: Upper Port, MP: Middle Port, and $n$ is the harmonic rank.

### 3. Simulation Setup of the Parallel SOA-MZIs System

The simulation setup dependent on the linkage among three parallel SOA-MZIs carried out by applying a Virtual Photonics Inc. (VPI) simulator [10,11] used to measure conversion gains (CGs) and error vector magnitude (EVMs) as displayed in Figure 2. In that transmission system, we also introduced an optical pulse clock (OPC) source [2] at the sampling frequency $f_s$ = 19.5 GHz at the wavelength $\lambda_s$ = 1550 nm that gives a 2 ps optical pulse train. The OPC used as a sampling signal, has harmonics at frequencies $nf_s$. The identical OPCs are used at the middle port (MP) for each SOA-MZI.

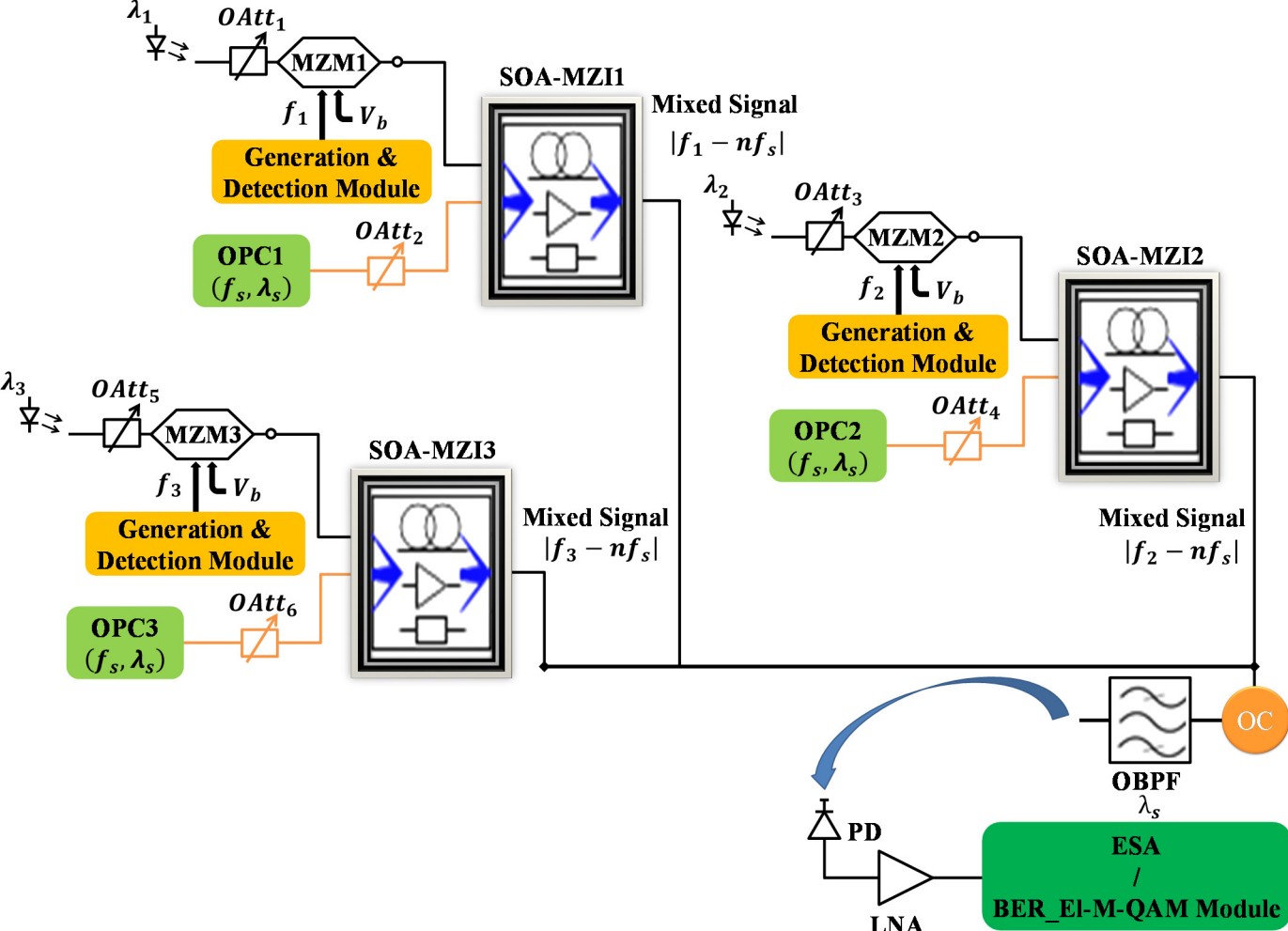

**Figure 2.** Simulation setup for CG and EVM calculations. OAtt: Optical Attenuator. MZM: Mach-Zehnder Modulator. PC: Polarization Controller. OBPF: Optical Band Pass Filter. LNA: Low Noise Amplifier. ESA: Electrical Spectrum Analyzer, OPC: Optical Pulse Clock. BER: Bit Error rate. QAM: Quadrature Amplitude Modulation.

The power level of harmonics of the sampling signal at the SOA-MZI input mainly depends on three parameters: the pulse shape, the pulse width, and the rank of the considered harmonic $n$. The technique encompassed in sampling SOA-MZI mixers is based on the convolution of an input RF signal with harmonics of the sampling signal. In order to keep the sufficient efficiency of the optical system relying on the sampling parallel SOA-MZIs link, the amplitude of harmonics is a significant factor, which has to be addressed. The pulse width is defined as the full width at half maximum (FWHM) measured at half of the maximum optical pulse power. The real pulse used at the SOA-MZI input corresponds to the Gaussian pulse train, thanks to its smoother shape, is closer to a genuine pulse. The harmonics of the Gaussian pulse train monotonically decline with the increasing of the

harmonic rank $n$. The pulse width plays on promptness that the power level of harmonics alters with $n$. It means that an output sampled signal of a sampling SOA-MZI mixer including a harmonic of a given rank must be driven by a sampling signal at the SOA-MZI input with a sufficient minimum pulse width to preserve an efficient performance through conversion gains (CGs) of sampling operation for this requested harmonic. The pulse width is 2 ps, which corresponds to a low duty cycle, which is the multiplication of the pulse width with the sampling frequency $(2 * 10^{-12} * 19.5 * 10^9 = 3.9\%)$, of the sampling signal. As a result, shorter pulses lead to lower alterations of the amplitude of harmonics with $n$, that is why only 2 ps pulses are used in this study.

The data signal is generated by a laser source, which is intensity modulated through an optical Mach-Zehnder modulator (MZM) driven by the generation and detection module at frequencies $f_m$. This module is used to establish the orthogonal frequency division multiplexing (OFDM) data at $f_m$. The optical MZM used at the upper port (UP) of each SOA-MZI has same characteristics and operating point. Besides, its parameters are given: the extinction ratio (*ER*) of the optical MZM is equal to 32.3 dB, the $-3$ dB bandwidth of the used MZM is 6 GHz, and the electrical power injected into the RF input of the MZM is 0 dBm. The RF signals are then down-converted to $|f_m - nf_s|$. The RF signals have the same optical power of $-10$ dBm at wavelengths ranging from $\lambda_1 = 1445$ to $\lambda_2 = 1446$ nm, then to $\lambda_3 = 1447$ nm. Besides, some physical parameters of the SOAs in the MZI are given: The bias current of SOA1 and SOA2 for each SOA-MZI is 350 mA, the SOA1 gain is 26 dB, the SOA2 gain is 28 dB, differential carrier lifetime is 24 ps, and SOAs saturation power is 15 dBm.

The sampled signals at each SOA-MZI output are combined to achieve simultaneous frequency down-conversion. This combined signal is optically filtered by an optical band pass filter (OBPF) tuned at $\lambda_s$ = 1550 nm. The OBPF bandwidth is nominated to be 0.65 nm. The filtered signal is subsequently photo-detected by a 100 GHz photodiode (PD) whose responsivity is 0.86 A/W. Then, the electrical signal is amplified by a 33 dB low noise amplifier (LNA). Finally, the BER_EL-M-QAM module is used to measure the bit error rate (BER). Then, the EVMs are obtained from BER values [12].

The photodiode (PD) is used to convert the optical signal to an electrical one. The RF signal at SOA-MZI input is 99 GHz. This signal is down-converted to lower frequencies starting from 79.5 GHz and ending in 1.5 GHz. Hence, a high bandwidth of the PD is required in order to achieve a photo-detection. Besides, the performance of the optical transmission system based on the three parallel SOA-MZI link is evaluated using electrical performance indexes, such as, the electrical conversion gain (CG) and EVM values. On the other hand, uni-travelling carrier (UTC) PDs that have high-power are being expounded in an accretion of applications such as RoF (radio over fibre), microwave, and antenna and radar systems [13–15]. UTC-PDs can be employed for frequency up- and down-conversion for the high level of the frequency range up to 300 GHz.

The dynamic characteristic of the SOA-MZI [7] depends on the carrier lifetime $(\tau_c)$ and the stimulated carrier recombination time $(\tau_s)$. According to the used SOAs, the corresponding effective carrier lifetime $(\tau_e)$ ranges from a few tens to a few hundred ps as defined in Equation (1) [16].

$$\tau_e = \frac{\tau_c \tau_s}{\tau_c + \tau_s} \tag{1}$$

In order to ameliorate the SOA-MZI dynamic characteristic, a SOA has to be biased with a high bias current and high optical powers have to be entered at its input. In our case, the bias current of both SOAs for each SOA-MZI is 350 mA, the peak power of the sampling signal is $-0$ dBm, and the RF signal power is $-10$ dBm.

The carrier densities of both SOAs are influenced by injecting the sampling signal, which modulates the data signal phase at the upper arms to generate a phase modulation. Hence, the broadened converted pulse is canceled due to the slow SOA recovery time. This switching configuration results in inspiring a transmission window, which can be lowered by using a differential configuration [17]. The performance of the frequency mixing

based on the parallel SOA-MZIs by the sampling technique used in various applications, especially radio over fibre (RoF) is studied and achieved excellent results.

## 4. Simulation Down-Conversion Results

The unparalleled performance of the simultaneous down-converted signals based on the parallel SOA-MZIs link is assessed by conversion gains (CGs). CGs are defined as a difference between electrical powers in dBm of the combined signal at $|f_m - nf_s|$ and the data RF signals at the SOA-MZIs input at $f_m$. The conversion gain (CG) reduces with the mixing frequency for the combined signal as illustrated in Figure 3. It reached 31, 29, and 25 dB at $|f_1 - 5f_s|$, $|f_2 - 5f_s|$, and $|f_3 - 5f_s|$ linked to the fifth harmonic of the sampling signal. Moreover, the difference in CGs between the down-conversion from 98 to 0.5 GHz related to the first RF signal and from 99 to 1.5 GHz pertaining to the third RF signal is 6 dB. The conversion gains (CGs) propound a low-pass behaviour whose cut-off frequency $f_c$ occurs at $f_c = 1/2\pi\tau_d$ [2,16], where $\tau_d$ is the differential carrier lifetime.

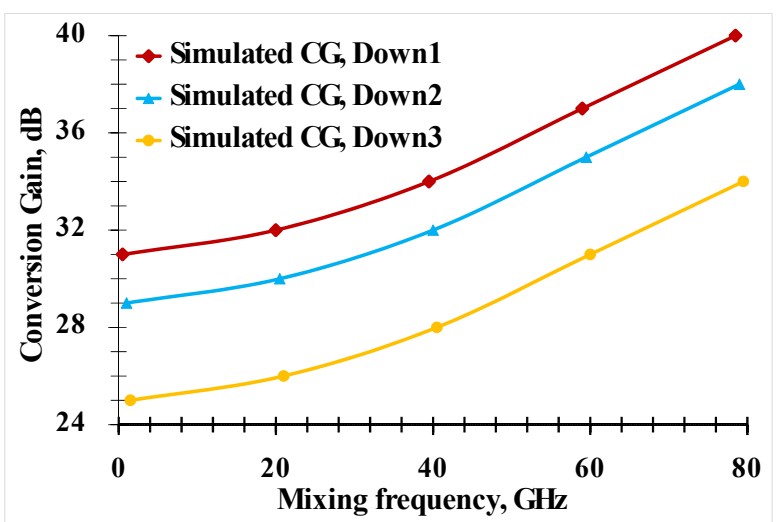

**Figure 3.** CGs of the simultaneous down-converted signal at the parallel SOA-MZIs output.

The quality of the simultaneous down-converted signals based on the parallel SOA-MZIs is evaluated by measuring EVMs [12]. The orthogonal frequency division multiplexing (OFDM) characteristics for 64 sub-carriers generated by the detection and generation module at the electrical port of the optical MZMs with a cyclic prefix (CP) of 25% are demonstrated in [18]. Figure 4 shows the EVM results of the sampled signal at $|f_m - nf_s|$ at the bit rate of 2 Gbit/s at the parallel SOA-MZIs output after combination. The EVM for OFDM modulation augments with the decrease of the mixing frequency. It reaches 29% at $|f_3 - 5f_s| = 1.5$ GHz related to the fifth harmonic of the sampling signal. This analysis is compliant with the conversion gain (CG) measurements shown in Figure 3, where the CG of the mixed OFDM signal at 1.5 can be evaluated up to 25 dB. Hence, the decrease of the CG of the mixed OFDM signal with the mixing frequency leads to the augmentation of EVM values.

It is worth noting that the EVM is calculated from the bit error rate (BER). The BER_EL-mQAM module is related to the BER, and the acceptable limit is defined as the value that provides a tantamount BER of 0.0038, which secures an error-free accomplishment after performing forward error correction (FEC) methods [19]. Hence, the QPSK acceptable limit is 35%. The FEC can considerably enhance the effectiveness of most applicable systems [20], especially radio over fibre systems. As a result, the 29% value is below the FEC boundary for the BER. On the other hand, the EVM requirement for realistic scenarios according to the current telecom standards is 17.5% [21,22] of the QPSK limit.

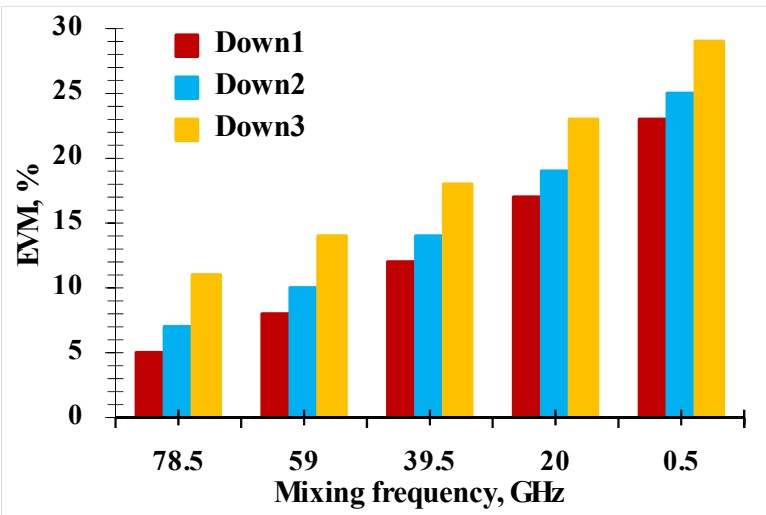

**Figure 4.** EVM values versus the mixing frequency for the combined down-converted signals at $|f_m - 5f_s|$ at the bit rate of 2 Gbit/s at the parallel SOA-MZIs output.

The signal to noise (SNR) is explained as a difference in dB between the electrical powers in dBm of the down-converted signal to the noise powers in dBm. It is only obtained at a variety of mixing frequencies ranging from $f_2 - f_m$ to $5f_2 - f_m$ to evaluate the performance of the optical transmission system as displayed in Figure 5. It decreases with the mixing frequency due to decreasing in the electrical power and the rising of the noise power related to the down-converted signals at $|f_m - nf_s|$. It ranges from 72 dB at $f_1 - f_s = 79.5$ GHz to 44 dB at $f_1 - 5f_s = 0.5$ GHz for the first down-converted signal. Furthermore, the SNR degrades about 8 dB over the entire mixing frequency range for the third down-converted signal compared to the first one.

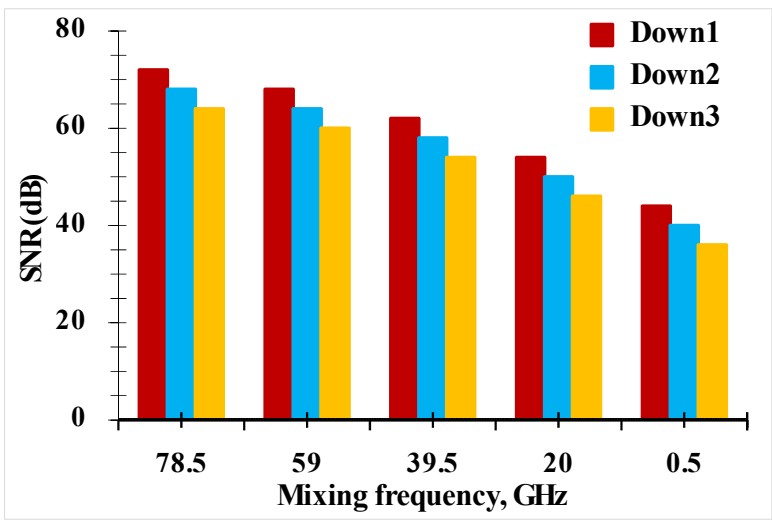

**Figure 5.** SNR of simultaneous down-converted signals for different mixing frequencies $|f_m - nf_s|$.

Previously, a format wavelength conversion system using parallelized SOA-MZIs is proposed [3–6]. In this work, we have used the three parallel SOA-MZIs link in order to achieve frequency mixing with better performance of the optical transmission system. As can be seen in Table 1, through conversion gain (CG), frequency range, and EVM, one can observe a big difference in the system models and their functions as well as the structure of the SOA-MZIs between our system and the work seen in [3–6]. Besides, the sampling method allows our system to reach a frequency range up to 99 GHz with excellent CG and EVM values. On the other hand, we have also compared this system with our previous

designs based on a single SOA-MZI [8] and a cascaded SOA-MZIs link [7] as given in Table 1. Our system for making the down-converted signal has been explored to upgrade the RoF (radio over fibre) system performance and minimize the system cost.

**Table 1.** Comparison of a variety of the optical transmission systems.

|  | Three Parallel SOA-MZIs | Two Parallel SOA-MZIs [3–6] | Two Cascaded SOA-MZIs [7] | Single SOA-MZI [8] |
|---|---|---|---|---|
| Format Modulation | QPSK | QPSK | QPSK | QPSK |
| CG (dB) | 25 | No data | 8 | 32 |
| Frequency Range (GHz) | 99 | No data | 86.3 | 80 |
| EVM (%) | 29 | 20 [5] | 18 | 15 |
| Bit rate (Gb/s) | 2 | 21.4 [3] | 40.5 | 8 |
| Up and Down Conversion | Down | Wavelength Conversion | Up and Down | Down |
| Technique | Sampling | Nonlinear Mixing | Sampling | Sampling |
| All-Optical | Yes | Yes | Yes | Yes |

## 5. Conclusions

In this study, we exhibit simulation analysis of the performance of the simultaneous down-conversion principle based on the three parallel SOA-MZIs link by using a band pass sampling method. The SOA-MZIs have the same structures, operating point, characteristics, and functions. Positive conversion gains (CGs) are achieved for the sampled signals at the sampling parallel SOA-MZIs output. The CG of 25 dB is attained for the third down-converted signal at 1.5 GHz correlated to the fifth harmonic of the sampling signal. The down-conversion of complex-modulated OFDM data signals has also been realized by the parallel SOA-MZIs system. The top EVM value at a bit rate of 2 Gbit/s is 29% for the third OFDM sampled signal at 1.5 GHz. This parallel SOA-MZIs system, because of its unmatched achievements, attracts attention for a variety of applications.

**Author Contributions:** All the authors, H.T. and A.M., contributed equally to this paper. All authors have read and agreed to the published version of the manuscript.

**Funding:** This research received no external funding.

**Institutional Review Board Statement:** Not applicable.

**Informed Consent Statement:** Not applicable.

**Data Availability Statement:** Data underlying the results presented in this paper are not publicly available at this time but may be obtained from the authors upon reasonable request.

**Conflicts of Interest:** The authors declare no conflict of interest.

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
