# Peer review of "Sampling Parallel SOA-MZIs Configuration for All-Optical Simultaneous Frequency Down-Conversion"

_photonics, doi:10.3390/photonics9100745_

Round 1

Reviewer 1 Report

A parallel sampling SOA-MZIs system to contrive simultaneous frequency down-conversion of three radio frequency (RF) signals based on a standard modulation configuration was proposed. The paper is well written. Therefore, minor revision is required. The suggestions for the authors are as follows.

The authors should give more parameters for the simulation, such as the extinction ratio of the MZM.

Author Response

Dear Professor,

Kindly, find attached, my response report.

Best regards,

Hassan Termos

Reviewer 2 Report

In this paper, a modulation concept to contrive simultaneous frequency down-conversion based on a three parallel SOA-MZIs link by using a band pass sampling method was proposed. The following problems need to be solved before accepted.

 it is better to give a performance comparison among this paper and the state of the arts using a table.

Author Response

(The authors gave the same response as above.)

Reviewer 3 Report

The paper illustrates a sampling approach for all-optical frequency down-conversion based on parallel SOA-MZIs. The scheme achieves positive conversion gains (CGs) for the conversion of complex-modulated OFDM data signals with the parallel SOA-MZIs system. The manuscript still needs some revisions to make it more promising, provided the following comments:

1.      The EVM of the down-converted signals goes up to 29% at the bit rate of 2Gbit/s, the authors should address whether it is applicable for practical system.

2.      It is true that SOA-MZIs provides high gain for the down-conversion. Could the authors comment on the signal-to-noise ratio or the signal quality after conversion?

3.      The PD used in the simulation features 100 GHz bandwidth and 0.86 A/W, which is far from feasibility.

4.      The authors should address how the pulse shape is affected by the carrier recovery of the SOAs after conversion.

Author Response

(The authors gave the same response as above.)

Reviewer 4 Report

In this work, results of simulations of down-conversion based on the three parallel SOA-MZIs link by using a sampling method are presented.

The work is interesting and clearly presented.

The subject matter discussed is on the one hand topical for many applications, but on the other hand quite unique and young, as relatively few centers deal with this topic.

I noticed some shortcomings in the work, the more important ones being:

1.    There are a lot of abbreviations in the text, it slows down the understanding of the paper. It is recommended to use only standard, generally accepted abbreviations (for example, MZM, MZI, BER etc.).

The abbreviation SOA-MZI should be explained in the abstract. Or abbreviations should be avoided in an abstract unless a term is used multiple times.

2.    Link in the 10th citation leads to the main page of the site, but not to the mentioned manual, it is difficult to find it. Please edit the hyperlink to the manual

3.    The choice of the factor n in the formula |fi-fs| needs for justifying. there are values n=1,5,10 in the text and on the figure 1 A-C

4.    Description of the Figure 1, that is the essence of the scheme simulation, has to be more detailed. Figure 1 (lines 99-100) shows successive signals - first a single one, then 4 identical ones and the last signal with a changed color on the right side. What does the color of frequencies (green, orange, etc.) mean? Why are heights of the left and right satellites of main (red) signal different?

Author Response

(The authors gave the same response as above.)

Round 2

Reviewer 3 Report

After revision, I would recommend the manuscript to be accepted for publication.